# A Preliminary Approximation to Microbiological Beach Sand Quality along the Coast of the Department of Atlántico (Caribbean Sea of Colombia): Influence of the Magdalena River

Hernando José Bolívar-Anillo [1], Zamira E. Soto-Varela [1,*], Hernando Sánchez Moreno [1], Diego Andrés Villate Daza [2], David Rosado-Porto [1,3], Shersy Vega Benites [1], Camila Pichón González [1] and Giorgio Anfuso [4,*]

[1] Faculty of Basic and Biomedical Sciences, Simón Bolívar University, Barranquilla 080002, Colombia
[2] Pacific Oceanographic & Hydrographic Research Centre, Tumaco 528501, Colombia
[3] Institute of Applied Microbiology, Justus Liebig University, 35392 Giessen, Germany
[4] Department of Earth Sciences, Faculty of Marine and Environmental Sciences, University of Cádiz, Polígono Río San Pedro s/n, 11510 Puerto Real, Spain
* Correspondence: zamira.soto@unisimon.edu.co (Z.E.S.-V.); giorgio.anfuso@uca.es (G.A.)

**Abstract:** Beaches represent important economic resources linked to "Sun, Sea and Sand" tourism and, therefore, their water quality constitutes an issue of great relevance especially in developing countries. The main objective of this work was to determine the microbial quality of beach sediments along the Caribbean coast of the Department of Atlántico (Colombia) and its relationships with the existence of local sources of contamination (e.g., streams containing wastewaters), beach exposition to waves, the quantity of beach visitors—which is reflected by beach typology (e.g., urban, rural, etc.), the presence of tourist activities/infrastructures and the beach sand sedimentological characteristics. Along the study area, samples of beach sediments were gathered in beach face and backshore areas at 11 sectors and the microbiological counts of three faecal indicator bacteria, i.e., *Escherichia coli*, *Enterococcus* spp. and *Clostridium perfringens*, were determined. A homogeneous distribution was recorded along the coast of *Escherichia coli* and *Clostridium perfringens* in both beach face and backshore sediments, in the order of 5 and 2 log CFU/100 mL, respectively; *Enterococcus* spp. was, at places, not observed in backshore sediments. No relationships existed between, on one side, the counts of the faecal indicator bacteria considered and, on the other side, the presence of streams and tourist activities/infrastructures, beach typology, exposition to waves and the sedimentological characteristics of beach sands. Such results suggest a chronic contamination of beach face and backshore sediments linked to the heavily polluted sedimentological load of the Magdalena River.

**Keywords:** *E. coli*; *Enterococcus* spp.; *C. perfringens*; beach face sediments; backshore sediments; beach typology

## 1. Introduction

Despite the decline recorded during the COVID-19 pandemic event [1], in the 21st century tourism represented complex and diverse activity of huge relevance [2,3]. In 2019, tourism generated 10.3% of global GDP, and coastal and maritime tourism represented the largest segment of this industry [4]. Coastlines with long stretches of clean, attractive sandy beaches have become relevant touristic economic zones with an estimated contribution in 2021 of USD 1.9 trillion, still well below the pre-pandemic value (USD 3.5 trillion) [5,6].

The selection of a beach by tourists is based on different factors, essentially water quality, safety, beach facilities, absence of litter and scenic beauty [7–9]. In the case of Colombia, especially during the last 15 years, the tourism sector recorded an incipient stage of development [10]. In 2021, tourism increased by 59.5% compared to 2020, generating a

contribution of ca. USD 3.1 million in foreign exchange to the country economy [11]. The Colombian Caribbean coast has a great potential for "Sun, Sea and Sand" (3S) tourism for its exuberant natural beauty and richness [12]. However, beaches close to Barranquilla, in the Department of Atlántico, have not adequate environmental conditions to offer appropriate touristic services, especially due to the presence of huge amounts of beach litter and vegetation debris, and microbiological contamination problems that may affect the health of local visitors and tourists [13–15].

It is estimated at a global scale that, annually, bathing in recreational coastal waters contaminated with faecal matter causes more than 120 million cases of gastrointestinal illness and 50 million cases of severe respiratory illness [16,17]. The main sources of recreational water pollution are considered to be poor functioning sewage treatment plants, septic tank systems, agricultural runoff from livestock farming, wildlife (e.g., migratory wildfowl), domestic animals and stormwaters [18,19]. In Latin America and the Caribbean, the most common source associated with water pollution on beaches is the uncontrolled discharge of untreated domestic wastewaters, which contain a high load of microorganisms of faecal origin causing diseases such as cholera, amebiasis, hepatitis, typhoid fever, paratyphoid, etc. [20]. Those bacteria can be transmitted from beaches to humans through physical contact with water and/or sand during recreational activities [21].

Historically, the assessment of the microbiological quality of seawater has focused on the culture and enumeration of faecal indicator bacteria (FIB) suspended in the water column [21]. International organizations such as the World Health Organization—WHO, the US Environmental Protection Agency and the European Community Commission have recommended the use of *Escherichia coli* and enterococci as suitable indicators to measure the microbiological quality of the water column [18,22], according to different environmental standards such as the ones established by the Council of Europe (2006/7/EC) [23] and the United States Environmental Protection Agency (Recreational Water Quality Criteria, 2012) [24]. In Colombia, the quality of seawater is regulated by the decree No. 1594 of 1984 (and modifications, i.e., decree No. 3930 of 2010) that has not been updated according to the new available techniques of microbiological analysis [13].

Such regulations concern only seawater quality and several different studies have found significantly higher concentrations of faecal indicator bacteria in sediments ($10^1$ to $10^6$ CFU or MPN/100 g wet weight for sediment). Therefore, sediments constitute large potential reservoirs of bacteria and their resuspension, linked to different natural processes (e.g., wave action) and human activities (e.g., bathing), represents an important source of delivering pathogens [18,22,25] that may significantly affect seawater quality assessments required by existing regulations [18].

Despite the above, there is presently no worldwide regulation that determines the permitted content for these microorganisms, as in the water column [23,24]. Different authors [19,21,22,25] state that studies are needed to characterize the impact that sediment-bound bacteria have on water quality and the health of bathers in order to standardize sampling and quantification methods, and results such as those shown in this study seek to generate this type of discussion both in the scientific community and among environmental authorities of different countries.

Sediment-bound bacteria can survive and persist for longer periods (months) than those found planktonically in the water column, which can survive only a few days [18]. Sediments offer bacteria advantages because they serve as a site for attachment (biofilm formation), a source of organic substances and nutrients, and provide protection against environmental stresses such as sunlight and protozoan grazing [26].

Several studies have observed that, in aquatic environments, bacteria are often associated with fine, cohesive, sediment particles (<60 μm) [27]. Bacteria are initially attracted to sediments by London–van der Waals forces and, once they are close to the surface, they can use extracellular polymers to form a strong and permanent adhesion [27]. However, the survival of sediment-bound microorganisms is affected by biological factors (e.g., predation

and competition), environmental conditions (e.g., particle size, temperature, humidity, nutrients and sunlight) and physical forces (e.g., waves, currents and tides) [28,29].

Therefore, the number of FIB and the presence of sediment-bound pathogens could pose a threat to human health if sediments are resuspended or come into direct contact with skin, or are ingested or inhaled by bathers [22,30]. Thus, the sole assessment of the microbiological quality of the water column may underestimate the presence of sediment-bound bacteria that are potential pathogens and may pose a health risk to bathers [31]. The reconstruction of the dynamic of each indicator gives relevant information about the risk of exposition to pathogens during recreational activities of beachgoers and an approximation of its transference from sediments to the water column and vice versa.

The present study is the first dealing with the determination of counts of *E. coli* and *Enterococcus* spp., as indicators of faecal contamination, and *C. perfringens*, as an indicator of remote episodes of faecal contamination [32–34], in beach face and backshore sediments gathered at 11 natural and urban beaches located on the Colombian Caribbean coast of the Department of Atlántico (Figure 1). The aims of the paper are to show that sediments act as a possible reservoir of such microorganisms and, in addition, evidence relationships between microorganism counts and environmental and human factors as well as the influence of water and sedimentary supplies of the Magdalena River.

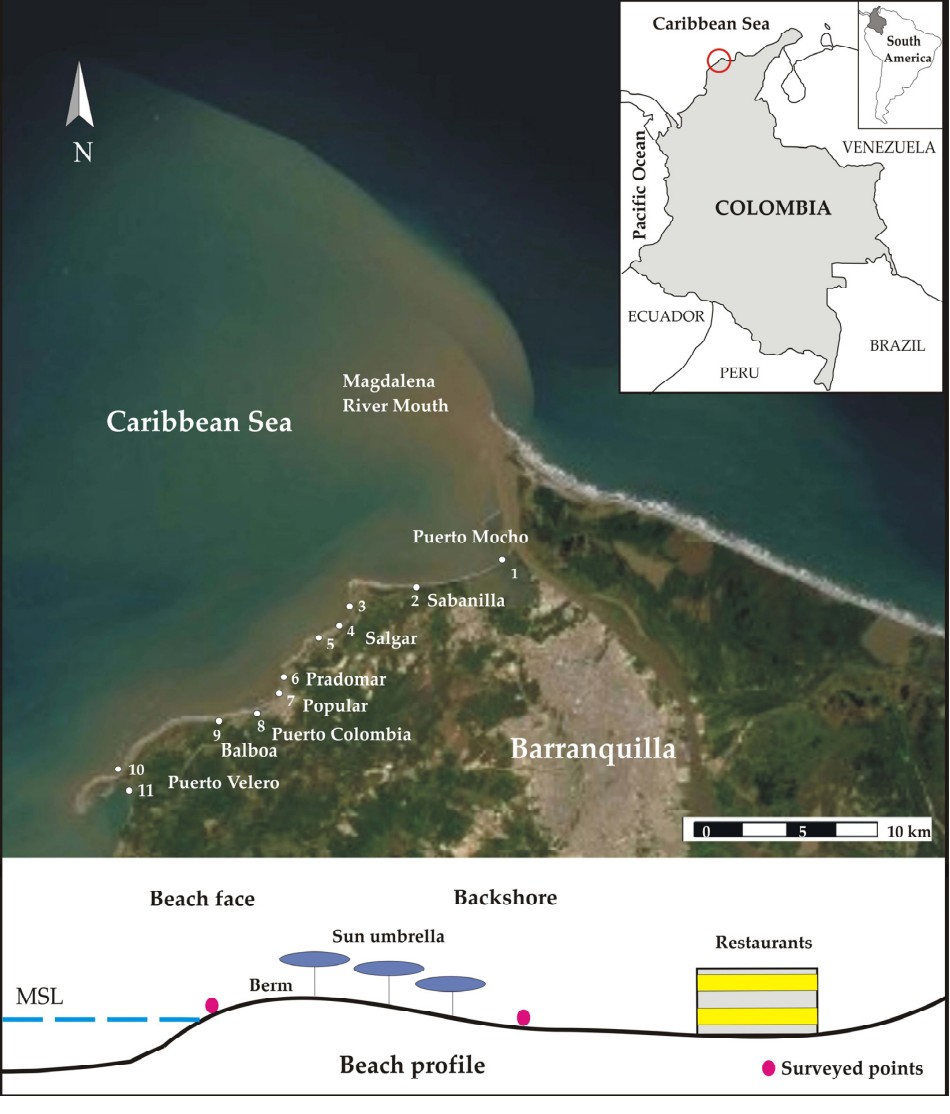

**Figure 1.** Location map of the sites investigated and position of sampled points along the beach profile.



The results of this study, which constitutes a preliminary investigation that needs to be enhanced in the future by means of deeper monitoring programs and the determination of the real health risk associated with the considered indicators, serve to enlarge the existing poor data base on beach sand quality in this zone and may be useful to update the microbial standards used by Colombian regulations concerning beach quality.

In addition, the information obtained, and the methodology used, can be easily applied to other coastal areas in South America or other continents with similar environmental characteristics/problems and, therefore, be useful for monitoring the microbiological quality of beaches to comply with international standards and ensure the safety of local bathers and national and international tourists.

## 2. Study Area

The study area, ca. 35 km in length, is located on the Caribbean coast of Colombia, in the Department of Atlántico, between the Magdalena River mouth and Puerto Velero (Figure 1). From an administrative point of view, the coastal zone of the Department of Atlántico includes the municipalities of Puerto Colombia, Tubará, Piojó, Juan de Acosta, Luruaco and Barranquilla [35]. The coastal municipality with the largest coastal area (46.77 out of 93 km$^2$) is Puerto Colombia, in the metropolitan area of Barranquilla. It has a population of 26,932 inhabitants and its economy is mainly based on tourism activities, especially the "3S" tourism [35,36], reflected by the presence of tourist activities/infrastructures at several sites (Figure 2).

The coastline is composed of quaternary sedimentary deposits shaped in complex and varied coastal features (e.g., sand spits, bars, lagoons and beaches) [37]. Beaches essentially show dissipative, flat profiles and relevant and frequent erosion and flooding events, i.e., due to the smooth beach slope wave run-up is able to penetrate tens of meters [38,39]. It is a tropical environment with seasonal variations in rainfall from the dry season (December–March) and the transitional seasonal (April–July) to the rainy season (August–November) [40]. Maximum annual precipitation is circa 2500 mm and mean annual temperature is ca. 27 °C [40]. Tidal range is mixed semi-diurnal, with maximum amplitudes of 60 cm, and trade winds (*Alisios*) are frequent during December–March [40]. Offshore waves approach from the first quadrant and secondary refracted and diffracted wave fronts approach the coast from the third and fourth quadrants; significant wave height, which is defined as the average wave height of the highest one-third of the recorded waves, is ca. 2 m and the average peak period is 7 s [41].

The net longshore drift has a dominant south-westward component; minor reversals to the northeast take place during the rainy period when southerly winds dominate and set up short, high-frequency waves [42,43].

Coastal morphology is essentially linked to the sedimentary supplies of the Magdalena River. The Magdalena River is the largest river system in Colombia with a length of 1612 km and a drainage basin covering 257,438 km$^2$ (24% of the Colombian territory) [44,45]. It starts in the Andes Mountains at an altitude of 3685 m and finally outflows into the Caribbean Sea, at Bocas de Cenizas, in the Department of Atlántico [44,45]. The Magdalena River forms a delta of 1690 km$^2$ that ends in a steeply sloping offshore canyon [44]. An amount of 724 municipalities in 19 of the 32 departments of the country are located in the Magdalena River basin, concentrating 80% of the total population of Colombia [45,46]. Sewerage coverage in those municipalities averages 81.81% in urban areas, while in rural areas it is only 17.31%. In Puerto Colombia, the largest municipality in the sampled area, only 55% of wastewaters are treated [47]. In addition, 334 (46%) out of 724 municipalities do not have a solid waste disposal system [46]. Therefore, the Magdalena River is considered as the main source of wastewater and litter pollution, and of a great quantity of vegetation debris for the beaches investigated [15,44,46,47] that often show huge amounts of leaves, tree branches, trunks and litter especially at sites close to the river mouth (e.g., Puerto Mocho and Sabanilla) and in remote and rural areas (e.g., Balboa

and Puerto Velero, and some sectors at Puerto Colombia) that have no beach clean-up programs (Figure 2).

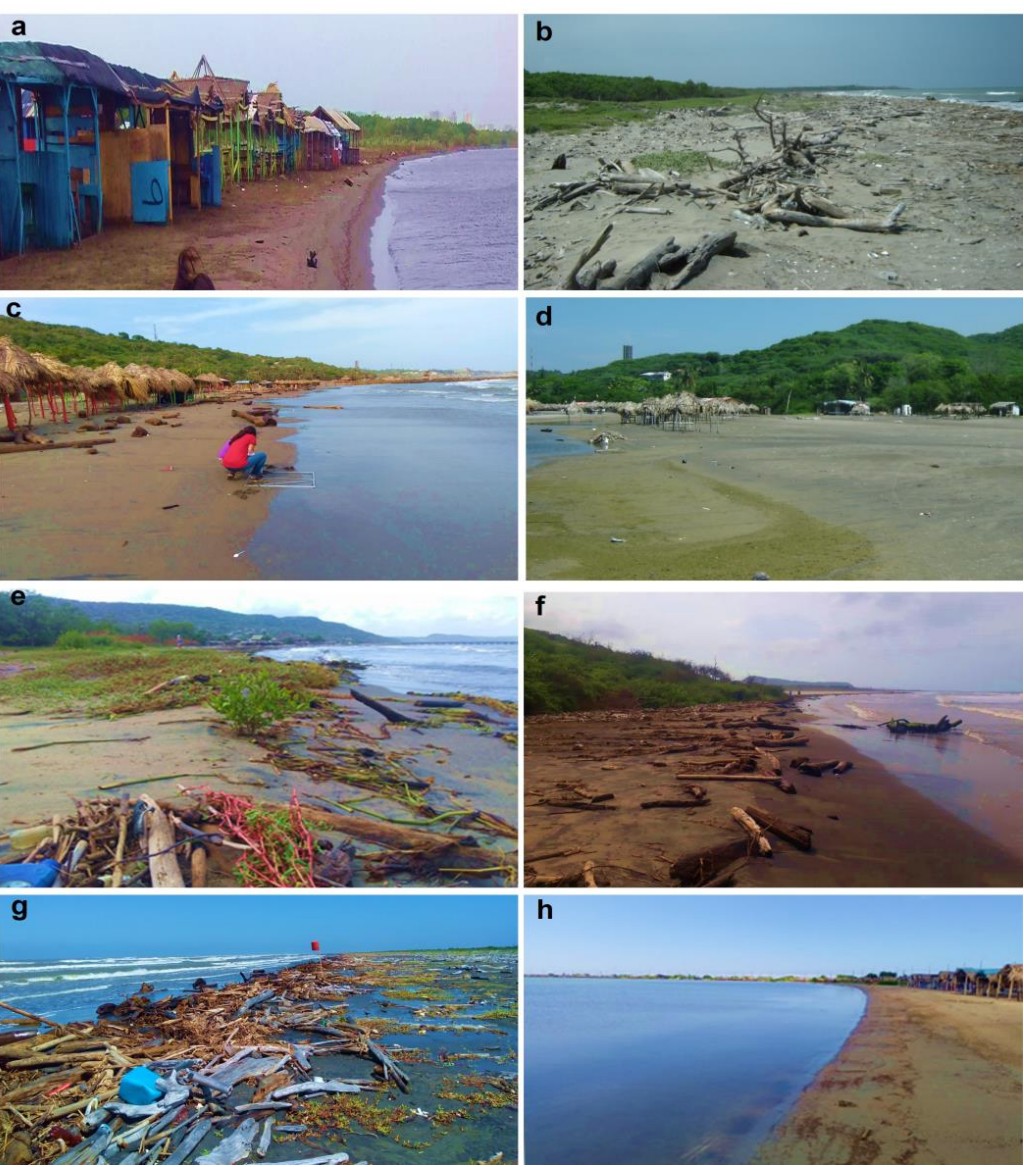

**Figure 2.** (**a**) Tourist activities/infrastructures at Puerto Mocho (Site 1, Figure 1). (**b**) Sabanilla beach densely covered by vegetation debris and beach litter (Site 2). (**c**) Sun umbrella at Salgar (Site 4); beach characteristics are very similar at sites 3 and 5. (**d**) Popular beach with a smooth profile and tourist activities/infrastructures (Site 7). (**e**) Puerto Colombia beach (Site 8). (**f**) Balboa beach with vegetation debris (Site 9). (**g**) Puerto Velero (Site 10), an exposed beach with a smooth profile and abundant vegetation debris and beach litter. (**h**) Puerto Velero (Site 11), a sheltered beach with a smooth beach profile and tourist activities/infrastructures (Figure 1).

The abundant quantity of organic matter (i.e., leaves, branches, trees, trunks, coconuts, etc.) have a major role in enhancing the biodiversity of the beach system and, along with stranded algae, constitute nutrients for foredune pioneer plants and dune bushes [15]. Beach litter and, especially, plastic materials, constitute contaminant agents because of their composition and adsorbed pollutants and have a negative impact on coastal scenery [4,8,46,47]. Both organic matter and litter deposited by the Magdalena River could act as reservoirs of FIB.

## 3. Materials and Methods

### 3.1. Marine Climate Characterization

The daily characteristics of the wind during the month in which the survey was carried out were obtained from the Network for the Measurement of Oceanographic Parameters and Marine Meteorology of the General Maritime Directorate of Colombia (MOPMM). Daily wave data for July 2018 were obtained from the Météo-France Global Ocean Reanalysis Wave System (WAVERYS) at $1/5°$ resolution [48]. The system provides an analysis of waves on the global ocean surface, based on a third-generation MFWAN model that calculates the wave spectrum, i.e., the distribution of wave energy. The significant wave height, which is defined as the average wave height of the highest one-third of the waves recorded, was used to characterize marine climate for the study area during the month of July 2018 and wave characteristics during the day of the survey at each sampled site.

### 3.2. Sand Sampling

In situ sampling was carried out on 9, 16 and 23 July 2018 and 33 sand beach samples were collected in natural and, especially, urban/tourist beaches. Sand samples were gathered in two beach sectors: the beach face (wet sand samples) and the backshore (dry sand samples, Figure 1). At each location, a quadrant of $1 \times 1$ m was randomly set and, at each corner, samples were gathered up to a depth of ca. 5 cm using a sterile spatula, obtaining a composite sample of 250 gr. Samples were stored in sterile Whirl-pack® sampling bags under refrigeration at 4 °C and transferred to the laboratory.

### 3.3. Microbial Analysis

The membrane filtration technique described in the 9–66 9222D method of Standard Methods for the Examination of Water and Wastewater [49] was implemented for the microbiological analysis of dry and wet samples. First, a portion of 10 g of sample was placed in a sterile Schott® flask with 90 mL of peptone water at 0.1%, and then it was hand shaken for two minutes and allowed to settle for 30 s [50]. The sample was decanted and the supernatant was used to carry out serial dilution up to $10^{-8}$. Afterward, each sample was filtered through a 0.45 μm pore size cellulose acetate filter by using Millipore® stainless steel equipment with 250 mL polysulfide filtration funnels; the filters were placed in the corresponding agar to each indicator.

The culture medium and incubation conditions were as follows: Concerning *E. coli* counts, they were determined using Agar Brilliance (Oxoid®) and the plates incubated at 37 °C for $21 \pm 3$ h. Purple colonies were considered positive to set the counts of this indicator. In the case of *Enterococcus* spp. quantification, Slanetz–Bartley agar (Oxoid®) was used and samples incubated at $44 \pm 1$ °C for $21 \pm 3$ h; red or pink colonies were considered positive to set the counts of this indicator. To determine *C. perfringens* counts, a Membrane *Clostridium* m-CP agar (Oxoid®) was used and the plates incubated in anaerobic conditions for $21 \pm 3$ h at $44 \pm 1$ °C. Only opaque yellow colonies were considered positive to set the counts of this indicator.

### 3.4. Granulometric Analysis

According to Giro and Maldonado [51], to determine the grain size of the analysed sediments, an amount of 200 g of sediment from each sample was sieved by means of a Ro-Tap machine during 15 min at 900 rpm according to the ISO 14688-1 standard, using sieves with 12.5, 4.75, 0.9, 0.6, 0.3, 0.18 and 0.075 mm mesh sizes, commonly used for soil classification purposes [52]. Grain size and statistical parameters were than determined according to the Udden–Wentworth scale [53] and Folk and Ward [54].

### 3.5. Statistical Data Analysis

Beach characterization data and microbiological results were analysed with R software version 1.3.1093 and Hmisc version 4.5-0. Measures of the central tendency and dispersion were calculated for each indicator counts. Spearman's rank-order correlation was

utilized to evaluate relationships among *E. coli*, *Enterococcus* spp. and *C. perfringens* counts, the presence of streams and tourist activities/infrastructures, and the sedimentological characteristic of the investigated beaches.

## 4. Results

### 4.1. Marine Climate Characterization

Wind speed at the investigated sites during the surveying days (9, 16 and 23 July 2018) ranged from 4.6 to 8.2 m/s (Table 1) and the predominant approaching direction was from the south–southeast. In relation to wave characteristics during the surveying days, the average significant wave height (H$_s$) at each sampled site was ca. 1.3 m, and the maximum wave height recorded 1.68 m and the minimum 0.20 m. Average wave period in the area was ca. 6.45 s (Table 1). Predominant wave fronts approached from the NNE with the exception of P10 Puerto Velero where the predominant direction was from the SW (Table 1). Last, the significant wave height trend for the whole investigated coastal area during July 2018 is presented in Figure A1 (Appendix A).

**Table 1.** Beach characteristics.

| Sampling Point and Location | Wind Speed (m/s) * | Wave Height (m) * | Direction (°) | Beach Exposition Level | Streams/ Outflows | Beach Typology | Tourist Activities/ Infrastructures |
|---|---|---|---|---|---|---|---|
| P1. PUERTO MOCHO | 8.2 | 0.54 | 010° | Low | Absent | Rural | Present |
| P2. SABANILLA | 8.2 | 1.43 | 010° | High | Absent | Rural | Present |
| P3. SALGAR | 8.2 | 1.26 | 010° | High | Absent | Urban | Present |
| P4. SALGAR | 8.2 | 1.26 | 010° | High | Absent | Urban | Present |
| P5. SALGAR | 7.4 | 1.68 | 010° | High | Present | Urban | Present |
| P6. PRADOMAR | 4.6 | 1.41 | 010° | High | Present | Urban | Present |
| P7. POPULAR | 7.4 | 1.43 | 010° | High | Present | Urban | Present |
| P8. PUERTO COLOMBIA | 4.6 | 1.41 | 010° | High | Present | Urban | Present |
| P9. BALBOA | 4.6 | 1.41 | 010° | High | Absent | Rural | Absent |
| P10. PUERTO VELERO | 7.4 | 1.68 | 010° | High | Absent | Remote | Absent |
| P11. PUERTO VELERO | 7.4 | 0.20 | 220° | Low | Present | Rural | Present |

Note(s): * Data corresponding to values observed on the day of the survey during July 2018 obtained from the Météo-France global ocean reanalysis wave system (WAVERYS).

### 4.2. Beach Characterization

The main characteristics of each sampled beach are described in Table 1. Beaches show flat profiles and backshore sediments present a certain level of humidity, and the beach phreatic level is often quite superficial [55]. Regarding their exposure to wave energy and wind, which affect beach dynamics and erosion rates, almost all sites were considered as exposed because affected by winds and wave fronts coming from the third and fourth quadrants. Puerto Mocho and Puerto Velero (Site 11) were classified as sheltered because respectively protected by the jetties at the Magdalena River mouth and the sand spit at Puerto Velero, as confirmed by wave data recorded during the study period (Figure 1 and Table 1).

Concerning freshwater supplies at investigated areas, streams outflowing directly onto the beach were identified at Salgar, Pradomar, Popular and Puerto Colombia; such streams are considered as a source of contamination of coastal waters [13]. Some streams are characterized by small continuous flows linked to a natural inland freshwater source, i.e., at Salgar, others are linked to natural freshwater flows and illegal pouring of wastewaters, i.e., at Pradomar and Popular, and the one at Puerto Colombia is linked to the local wastewater treatment plant that discharges inadequately treated water directly on the beach. Last, the discharge of local, poorly treated wastewaters into the seawater column is observed at Puerto Velero (Site 11) [56].

According to their location in urban areas or not, and their accessibility [57], three types of beach were differentiated: (i) urban, located at (or very close to) urban areas, i.e., Salgar (Sites 3–5), Pradomar and Puerto Colombia; (ii) rural, far from human settlements, i.e., Puerto Mocho, Sabanilla, Balboa and Puerto Velero (Site 11); (iii) remote, distant from urbanized areas and of difficult access, i.e., Puerto Velero (Site 10), Table 1.

Lastly, tourist activities/infrastructures, which correspond to bars, kiosks and restaurants placed in the middle of the backshore and/or at beach landward edge, are observed in most sites (82%), Figure 2. Tourist activities/infrastructures were not observed at Balboa and Puerto Velero (Site 10), which respectively correspond to remote and rural beaches.

### 4.3. Sedimentological Characteristics

The results of the granulometric analysis show almost all samples consisting of "Medium sand" with "Coarse sand" only observed in backshore sediments at P1, P3 and P5 (Table 2). Sorting values essentially correspond to "Moderately well sorted" sand with three cases of "Well sorted" and one of "Very well sorted" sands (Table 2).

**Table 2.** Sedimentological characteristics of investigated beaches.

| Sampling Point | Sample Location | Mean (phi) | | Sorting | |
|---|---|---|---|---|---|
| P1. PUERTO MOCHO | Beach face | 1.63 | Medium sand | 0.51 | Moderately well sorted |
| | Backshore | 0.56 | Coarse sand | 0.39 | Well sorted |
| P2. SABANILLA | Beach face | 1.59 | Medium sand | 0.53 | Moderately well sorted |
| | Backshore | 1.52 | Medium sand | 0.52 | Moderately well sorted |
| P3. SALGAR | Beach face | 1.57 | Medium sand | 0.50 | Well sorted |
| | Backshore | 0.55 | Coarse sand | 0.44 | Well sorted |
| P4. SALGAR | Beach face | 1.81 | Medium sand | 0.54 | Moderately well sorted |
| | Backshore | 1.50 | Medium sand | 0.51 | Moderately well sorted |
| P5. SALGAR | Beach face | 1.63 | Medium sand | 0.54 | Moderately well sorted |
| | Backshore | 0.96 | Coarse sand | 0.31 | Very well sorted |
| P6. PRADOMAR | Beach face | 1.50 | Medium sand | 0.57 | Moderately well sorted |
| | Backshore | 1.54 | Medium sand | 0.54 | Moderately well sorted |
| P7. POPULAR | Beach face | 1.86 | Medium sand | 0.51 | Moderately well sorted |
| | Backshore | 1.47 | Medium sand | 0.51 | Moderately well sorted |
| P8. PUERTO COLOMBIA | Beach face | 1.82 | Medium sand | 0.61 | Moderately well sorted |
| | Backshore | 1.52 | Medium sand | 0.57 | Moderately well sorted |
| P9. BALBOA | Beach face | 1.43 | Medium sand | 0.59 | Moderately well sorted |
| | Backshore | 1.45 | Medium sand | 0.66 | Moderately well sorted |
| P10. PUERTO VELERO | Beach face | 1.59 | Medium sand | 0.61 | Moderately well sorted |
| | Backshore | 1.59 | Medium sand | 0.49 | Well sorted |
| P11. PUERTO VELERO | Beach face | 1.48 | Medium sand | 0.62 | Moderately well sorted |
| | Backshore | 1.50 | Medium sand | 0.52 | Moderately well sorted |

Most important sediment fractions were "Medium" and "Fine" sands and, secondarily, "Coarse" sand, the "Very fine" sand fraction being almost null (Figure 3). Therefore, it is possible to state that no relevant differences in grain size characteristics are observed either in alongshore or in cross-shore directions.

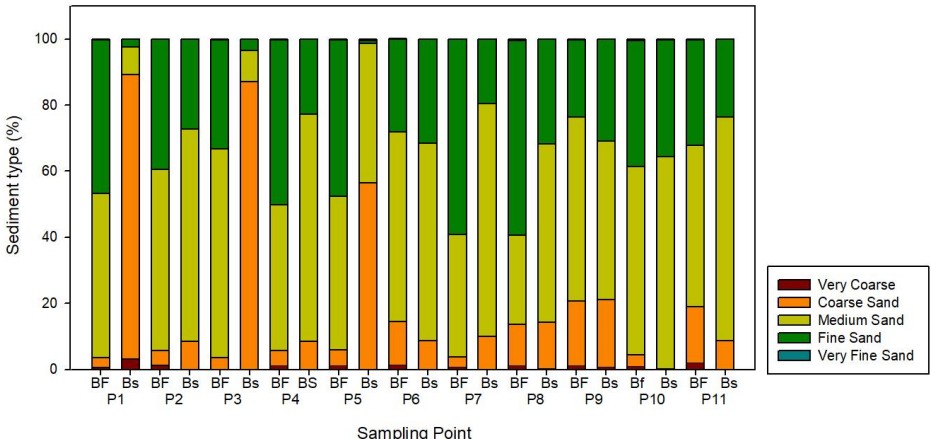

**Figure 3.** Sediment fractions of the studied samples, Bf: Beach face; Bs: Backshore.

*4.4. Distribution of Microbiological Indicators*

The general trend for the three indicators *E. coli*, *Enterococcus* spp. and *C. perfringens* are showed in Figure 4.

Concerning the counts of *E. coli*, it shows the highest values (with respect to the others) with similar average values observed in the beach face and backshore sediments, i.e., 4.15 and 3.95 Log CFU/100 g, respectively, and low variations for the median values that ranged from 4.12 Log CFU/100 g for the beach face sediments to 3.59 Log CFU/100 g for the backshore sediments. The standard deviation is more or less similar ranging from 0.64 for the beach face to 0.76 Log CFU/100 g for the backshore sediments (Figure 4a).

The recorded low spatial alongshore and cross-shore variations, from 3.5 to 5 Log CFU/100 g, are also observed looking at specific values recorded at each beach. The highest counts are recorded at P2 (Sabanilla), P10 and P11 (Puerto Velero), which do not present streams and show very similar values for beach face and backshore sediments. The lowest counts are observed at P9 (Balboa), P8 (Puerto Colombia) and P1 (Puerto Mocho) (Figure 4b).

In relation to *Enterococcus* spp., at most places, its counts record variations between beach face and backshore sediments, with an average and median value at the beach face of 2.51 and 2.92 Log CFU/100 g, respectively (Figure 4c); however, more than half of the backshore sediments does not show *Enterococcus* spp. counts (Figure 4d). The highest counts are recorded at P2 (Sabanilla) at the beach face, followed by the dry beach at P11 (Puerto Velero). Counts are higher in the backshore (in the order of 3 Log CFU/100 g, Figure 4d) than in the beach face at P6 (Pradomar) and P10 (Puerto Velero) and, especially, at P4 (Salgar) and P11 (Puerto Velero) (Figure 4d).

Finally, *C. perfringens* presents the lowest counts among the three evaluated indicators. The box plot shows few variations between the average value (1.99 Log UFC/100 g recorded in the beach face and 1.82 Log CFU/100 g recorded in the backshore) and the median value (1.89 Log CFU/100 g recorded in the beach face and 1.83 Log CFU/100 g recorded in the backshore) and likewise shows very small standard deviation values, i.e., 0.2 Log CFU/100 g for both cases (Figure 4e). At most places, the counts are very similar and in the order of two logarithmic units, such values being similar for the beach face and the backshore (Figure 4f).

*4.5. Microbiological Indicators versus Beach and Sedimentological Characteristics*

No relevant correlations are observed among microbiological counts and the presence of streams and tourist activities/infrastructures according to Spearman's rank-order correlation (P > 0.05). All beach characteristics shown in Table 1 are represented in a radial plot (Figure 5). Each vertex corresponds to the average value of each one of the nine parameters presented in Table 1. In the case of *E. coli*, the counts are similar for the different types of beach, in the order of 4 Log CFU/ 100 g for both the beach face and backshore sediments,

but the remote beach (P10, Puerto Velero) shows a value slightly higher (Figure 5a,b). The counts of *Enterococcus* spp. are mostly in the order of 3 Log CFU/100 g in the beach face sediments and 1 Log CFU/100 g in the backshore sediments. Concerning the beach face sediments, small variations are observed from place to place, the most relevant being the lowest counts recorded at sheltered beaches (Figure 5a). In the backshore sediments, *Enterococcus* spp. is less abundant than *C. perfringens* and shows the highest counts in remote beaches (Figure 5b). Finally, average counts of *C. perfringens* are similar at all beach types, in the order of 1 log CFU/100 g for both the beach face and backshore sediments, and are not related to any one of the considered variables.

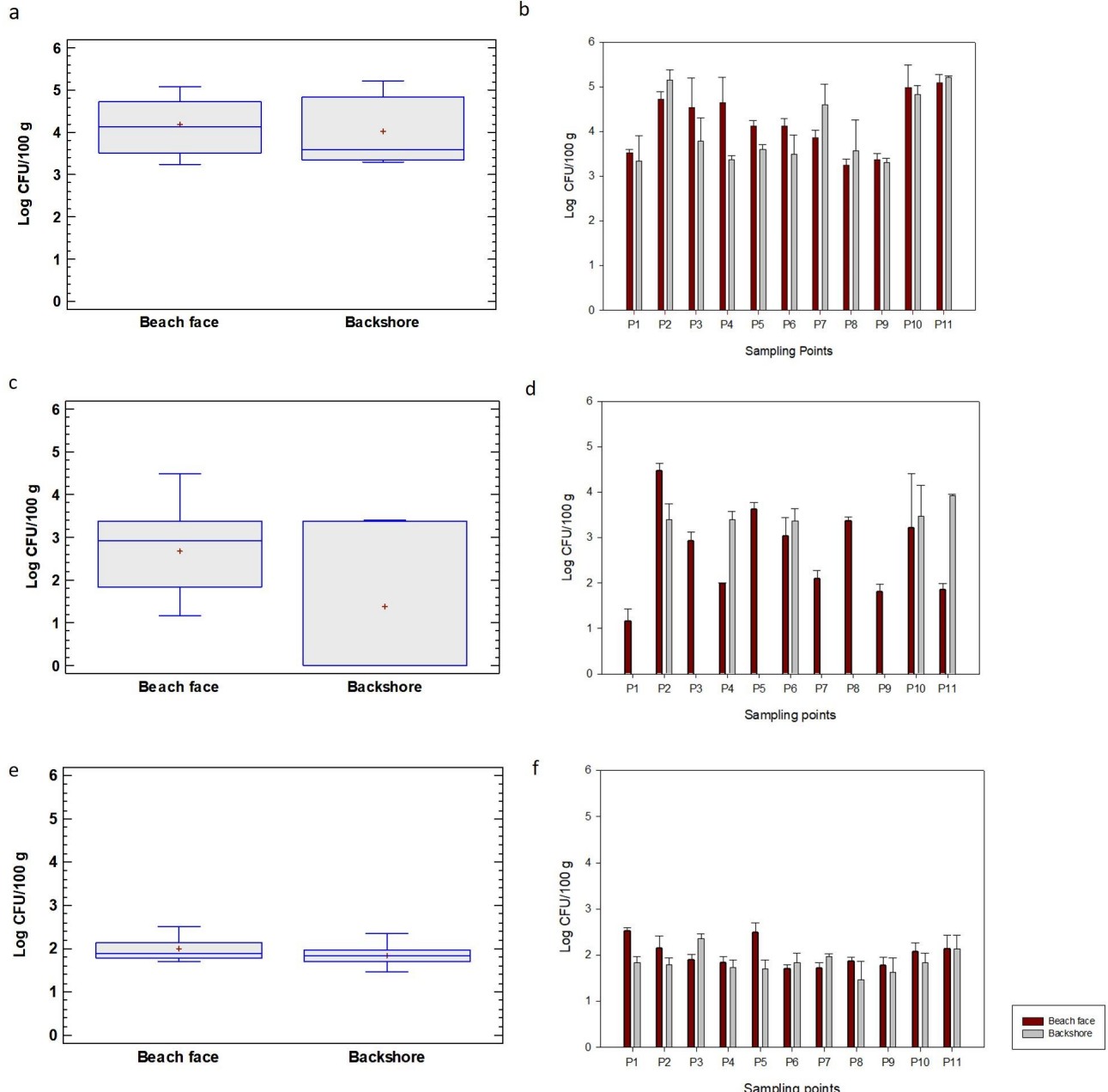

**Figure 4.** Microbiological results of the sampled sediments located in the beach face and the backshore. Boxplots show the counts of each indicator at the two studied beach areas and the observed alongshore distribution: (**a**,**b**) *E. coli*; (**c**,**d**) *Enterococcus* spp. and (**e**,**f**) *C. perfringens*. The vertical line inside the box represents the median, the red cross the average value and the limits of the whiskers the standard deviation.

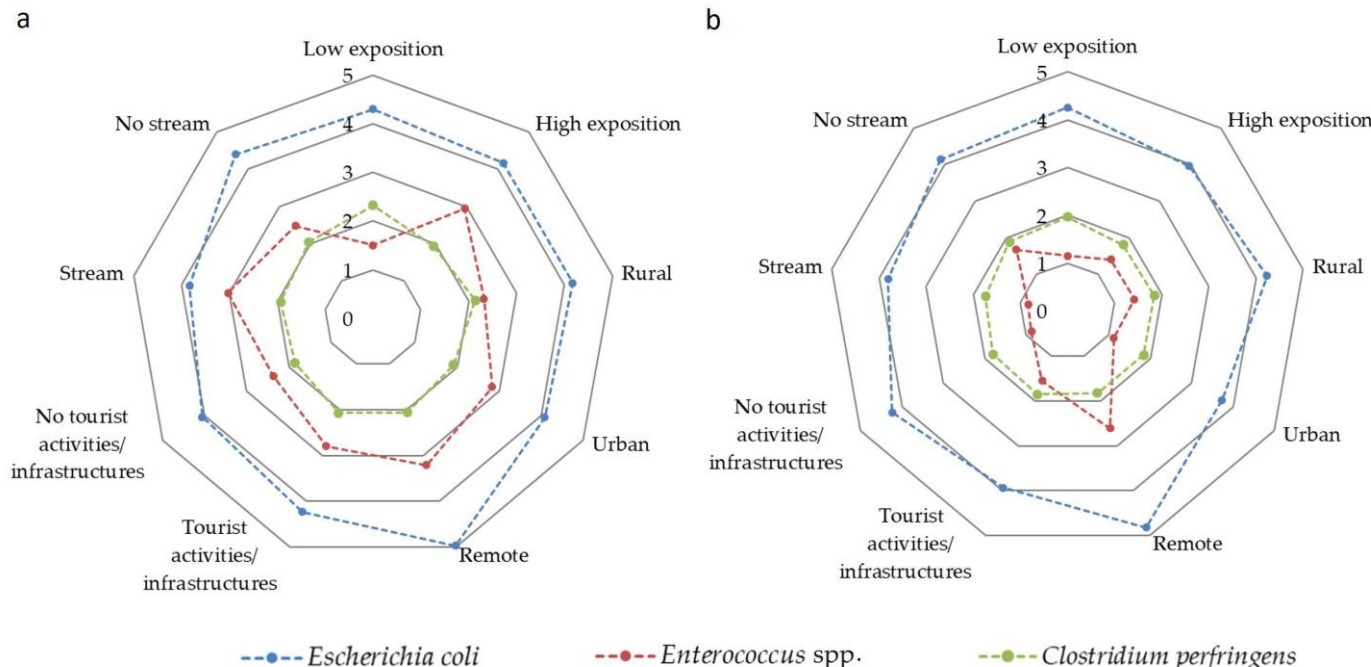

**Figure 5.** Microbial results according to beach characteristic presented in Table 1. (**a**) Beach face sediments and (**b**) backshore sediments.

Concerning the sedimentological characteristics, the results obtained in this paper show very small longshore and cross-shore variations and, in general, no apparent relationships with FIB counts recorded in both beach face and backshore sediments (Table 2, Figure 3). At places, the absence of *Enterococcus* spp. in the backshore sediments is accompanied by the presence of coarse and/or well and very well sorted sands, i.e., at P1, P3 and P5 (Table 2 and Figures 3 and 4), but such a trend is not confirmed for other backshore samples, i.e., P7, P8 and P9 (Figures 3 and 4) that only show very small differences between beach face and backshore sand characteristics (Table 2).

## 5. Discussion

The area investigated in this paper is close to the Magdalena River mouth (Figure 1); therefore, it is relevant to consider its influence in coastal waters and beach sediment pollution. According to different authors, in Latin America and the Caribbean, one of the main causes of coastal contamination is the discharge of untreated (or inadequately treated) wastewaters, which often contain a high load of faecal microorganisms [13,14,20,58,59]. As observed by Fernández-Nóvoa et al. [60], plumes originating at river mouths affect the physical, chemical and ecological dynamics of nearby coastal areas due to inputs of freshwater, dissolved nutrients, pollutants and suspended load. At the mouth of the Magdalena River, due to water salinity differences, is observed a well-defined and large river plume (easily visible in Figure 1) composed of freshwater floating on subjacent saltwater. The plume, which contains large amounts of sediments commonly observed near the top of the water column [61], is usually south-westward directed with an average velocity from 0.1 to 0.3 m s$^{-1}$ because of swell waves generated by trade winds blowing from the NE [44].

In terms of microbiological quality, the Magdalena River mouth is annually monitored by INVEMAR (the National Institute for Marine Research), who have detected high counts of microorganisms indicative of faecal contamination [62–64]. The last report, dated 2019, highlighted that the counts of thermotolerant coliforms (including *E. coli*) were ca. 676,000 MPN/100 mL, exceeding 3380 times the value accepted by the Colombian standard (200 MPN/100 mL) [65]. Different studies have reported that most enteric bacteria in

aquatic systems are bound to sediments and this binding influences their transport [13]. Bacteria are usually associated with silt and cohesive particles (<60 μm) [13] that are very abundant in the Magdalena River plume. Villanueva (2020) established that, at the mouth of the Magdalena River, 15.7% of the sediments correspond to sand-sized particles (63 μm to 2000 μm) and 84.3% to silt-sized particles (4 μm to 63 μm) [14].

Sediments arrive and are accumulated at investigated beaches, but sand-sized sediments are stable meanwhile silt-sized sediments are easily washed away since they are not stable in such environments (Table 2), as also observed by Gast et al. [66] in a study conducted at Kitty Hawk (USA). Furthermore, during the dry period, the saline wedge penetrates ca. 4 km upstream of the river mouth and, under such conditions, salinity at the previously mentioned river section is 6 ppt in the shallow layer and 33 ppt in the deeper layer, while in the deltaic front salinity values in the shallow and deep layers are 20 ppt and 35 ppt, respectively [15]. Suspended sediments in aquatic systems as well as bacteria are generally negatively charged, so they repel each other [13,16]. However, in the presence of high electrolyte concentrations, repulsive forces are suppressed and attractive forces (e.g., London–van der Waals) may result in reversible adsorption of bacteria to solid surfaces [13,17].

Several studies evidence that the Magdalena River is an important source of litter that ends up on the coast of the Department of Atlántico [46,67–69], along which 68% of the beaches are very polluted by the presence of thirteen different beach litter types composed of different plastics compounds, cloths, rubber, glass, wood, paper, etc. [44]. Those materials could serve as reservoirs for both FIB and pathogenic microorganisms [45,46]. In addition, a large number of plants (e.g., *Eichhornia crassipes*) and abundant plant debris transported by the Magdalena River also act as a source and reservoir of pathogenic bacteria, which are deposited on the investigated beaches especially in the rainy season. Therefore, the high level of pollution, the intrusion of the salt wedge and the huge abundance of suspended sediments in the Magdalena River could be key factors for the transport of faecal contamination indicator bacteria to beach sediments of the Department of Atlántico, that presented high counts of *E. coli* (5 Log CFU/100 g), *Enterococcus* spp. (up to 4 Log CFU/100 g) and *C. perfringens* (2 Log CFU/100 g). The relevance of the Magdalena River inputs of FIB versus the inputs of local streams is confirmed by the homogeneous alongshore distribution of the above-mentioned indicators (Figures 4 and 5). Further, counts and distribution of FIB are apparently neither affected by the presence of restaurants and kiosks, which discharge directly on the beach small amounts of wastewaters and organic matter, nor by the affluence of beach visitors that is directly related to beach typology (Figures 4 and 5).

No clear relationships are also observed among FIB and the level of beach exposure to waves, and beach face and backshore sedimentological characteristics (Figures 3 and 4). Thus, it is possible to state that the Magdalena River sediments constitute a chronic source of coastal contamination during most of the year, when the plume is south-westward directed, because faecal indicators are transported downdrift attached to sediments, and are transferred and progressively accumulated in beach sediments. Therefore, sediments show higher counts of indicator microorganisms than the water column, as previously established by Sánchez et al. [13]. Such a pattern has been described in several studies that have demonstrated the influence of sediments transported by rivers on beach microbiological quality. Huang et al. [70] developed an integrated hydro-bacterial model to simulate the adsorption–desorption processes of faecal bacteria to and from sediment particles in freshwaters of the Ribble River, and in estuarine and coastal waters of the Fylde coast (UK), and demonstrated that the resuspension of sediments increases FIB in bathing waters. Ferguson et al. [71] determined that the coastal waters of Huntington State Beach and Baby Beach (USA) may be receiving faecal indicator bacteria from the intertidal sediments of the Santa Ana River that contains high levels of FIB. The interaction between FIB and suspended sediments was also recorded at the study area by Torres-Bejarano et al. [56] that investigated the effects of beach tourists on bathing waters and sand quality at Puerto

Velero (Site 11) and found that the binding of bacteria of faecal origin to the sediments transported by the Magdalena River is a dynamic process.

Regarding the microbiological results of this study, the homogenous alongshore and cross-shore counts obtained at all beaches sampled—despite the existence or not of an evident source of local faecal contamination (e.g., a polluted stream or tourist activities/infrastructures) in the environment—has been also reported in other places [72].

The above can be due to the ability of FIB to form biofilms in different types of sediments that allow them to be transported to alternative sites (i.e., beaches) [73] and may result in their survival and favourable growth in beach face conditions due, in part, to the increased accessibility to nutrients [16,74,75] and the presence of organic material [75,76].

The high counts observed in this study in the backshore sediments may be partly attributable to a lower predation activity because of the lower water content with respect to the beach face environment [77]. Ishii et al. [78] determined that a low water potential in soil reduces the growth rate of *E. coli* but that survival rates are not different in dry and wet soils, which coincides with the results presented in this paper. In fact, in places, *Enterococcus* spp. counts are higher in the backshore than the beach face, this result being similar to other works that have recorded *Enterococcus* spp. counts higher than 1000 CFU.g$^{-1}$ in dry sands [16], and such counts being one to three orders of magnitude higher than the counts recorded in wet sands in USA [79] and in Colombian beaches [80]. The high backshore FIB counts may be also related to the fact that the beaches studied show dissipative and flat profiles and, therefore: (i) the phreatic level is close to the beach surface and thus even backshore sediments have a certain intergranular water content and (ii) due to the smooth beach slope, wave run-up is often able to penetrate tens of meters allowing the arrival of nutrients [16,77].

Furthermore, the sampling carried out in this study was conducted during early morning hours, i.e., under low sun irradiations conditions. This could favour the high counts of microorganisms recorded in the backshore as observed by Abdelzaher et al. [79] who associated the high observed levels of FIB in dry sands to different factors including the lack of sunlight during and immediately prior to sample collection. The heterogenic distribution of *Enterococcus* spp. counts could be linked, according to Bonilla et al. [77], to the fact that it has a very irregular or "patchy" distribution in beach sediments and this trend could be exacerbated in the backshore sediments (where this microorganism is not recorded at Puerto Mocho, Salgar, Popular, Puerto Colombia and Balboa, Figure 4) by the great germicidal effect of sunlight on these bacteria [81]. Additionally, differences observed at Puerto Mocho and Salgar could be related to the presence of coarse and well sorted sand (Table 2, Figure 3). This trend anyway is not observed at other sites and is difficult to confirm since the survival and persistence of enteric bacteria in coarse sands is an issue only addressed by a few studies [82]. Last, *Enterococcus* spp. counts are slightly lower in sheltered beaches too, which may demonstrate a lesser influence (with respect to exposed beaches) of wave action that brings contaminants and nutrients to the bacteria already present in the beach [70].

In this paper *C. perfringens* was also used as a third indicator of faecal contamination despite its determination not being required in any microbiological beach quality standard, but its ability to produce spores and its longevity [83] make it a valuable indicator of remote chronic faecal contamination [13,32,33,83,84]. This was also recorded by Kraus et al. [33] who established that *C. perfringens* has better survival capacity and greater retention in sediments than other microorganisms indicative of faecal contamination, since its spores persist in sediments for an undetermined time where other FIB decline more rapidly. Additionally, Lisle et al. [85], in a study conducted in McMurdo Station, Antarctica, established that *C. perfringens* concentrations in sediments were higher at greater distances from the sewage discharge site compared to counts of other FIB indicators, and Miller-Pierce and Rhoads [40] established that *C. perfringens* improves the detection of non-point-source contamination of marine waters; for those reasons it can be used as the main indicator bacteria for faecal contamination in tropical marine waters.

The presence of *C. perfringens* alone indicates that the lapse time between the exposure to faecal contamination and the sampling was longer than the decay rates for other indicators [85,86].

Last, considering that bathing in polluted waters causes approximately 120 million cases of gastrointestinal illnesses and 50 million cases of acute respiratory illnesses per year worldwide [17], high counts of the indicators evaluated in this study potentially pose a public health risk because of the shedding of bacteria from beach sediments to the water column. Some pathogenic strains of *E. coli* (e.g., O157:H7) have a low infectious dose and a small change in the number of these microorganisms in the water column can put the health of bathers at risk [87]. In addition, some strains of *E. coli* (including potentially pathogenic strains) are able to survive long periods of time and to reproduce in extraintestinal environments in tropical and subtropical areas, a phenomenon known as naturalization [73,88,89]. Those naturalized strains are indistinguishable from faecal-derived strains and may not indicate recent faecal contamination [72]. It is mandatory in future studies to establish if the bacteria observed in beach sands along the Department of Atlántico represent a real risk for beachgoers' health and if those bacteria suffered a naturalization process which would question their use as indicators of faecal contamination [46,75,90]. Last but not least, it has to be confirmed, by the application of molecular biology techniques, the relation between the bacteria recorded in beach sediments with ones observed in the Magdalena River.

## 6. Conclusions

This paper presents the results of a preliminary study on faecal indicator bacteria counts and their distribution in beach sediments along the Department of Atlántico, in the Caribbean Sea of Colombia. Such data were analysed according to the presence of tourist activities/infrastructures and local sources of faecal contamination (e.g., streams containing wastewaters) as well as beach typology, exposure of sites to wave action and grain size characteristics at beach face and backshore sediments. The influence of the Magdalena River water and sedimentary supplies was also taken into account.

The high counts of *E. coli*, *Enterococcus* spp. and *C. perfringens* recorded in beach sediments and their almost homogeneous alongshore and cross-shore distribution, and the absence of any relationships with the above-mentioned parameters, support the hypothesis that the Magdalena River, which has a relevant load of sediment-bound faecal bacteria, represents a chronic source of coastal contamination.

The absence of relevant differences between bacteria counts in beach face and backshore sediments may be related to the dissipative and flat profiles of the beaches investigated. They are easily flooded, allowing this way the periodic arrival of water and nutrients to the backshore. Further, they show a superficial phreatic level that allows the presence of intergranular water content in the backshore sands.

Regarding the path forward, in order to have a global view on beach sand quality, monitoring surveys have to be repeated in different seasons. Future investigations have also to determine, by the application of molecular biological techniques, if the high bacterial load recorded in beach sands along the Department of Atlántico represents a real risk for beachgoers' health, i.e., if such bacteria are (or are not) naturalized to beach environments and if they are genetically related to the faecal indicator bacteria of the Magdalena River.

**Author Contributions:** Conceptualization, H.J.B.-A., H.S.M., Z.E.S.-V., D.A.V.D., S.V.B. and G.A.; methodology, Z.E.S.-V., D.R.-P., H.J.B.-A. and C.P.G.; sampling, C.P.G. and H.S.M.; formal analysis, D.R.-P., Z.E.S.-V. and D.A.V.D.; investigation, H.J.B.-A., C.P.G., H.S.M., D.R.-P. and Z.E.S.-V.; data curation, D.A.V.D., D.R.-P., Z.E.S.-V. and G.A.; writing—original draft preparation, Z.E.S.-V., H.J.B.-A., D.A.V.D., S.V.B. and G.A.; writing—review and editing, Z.E.S.-V., D.A.V.D., H.J.B.-A. and G.A.; project administration, H.S.M., Z.E.S.-V. and H.J.B.-A. All authors have read and agreed to the published version of the manuscript.

**Funding:** This research was funded by the Simón Bolívar University.

**Data Availability Statement:** Not applicable.

**Acknowledgments:** The authors thank the Escuela Naval de Suboficiales ARC Barranquilla for granulometric analysis. This work is a contribution to the PROPLAYAS Network, the Andalusia (Spain) Research Group RNM-328 and the Center for Marine and Limnological Research of the Caribbean CICMAR (Barranquilla, Colombia).

**Conflicts of Interest:** The authors declare no conflict of interest.

**Appendix A**

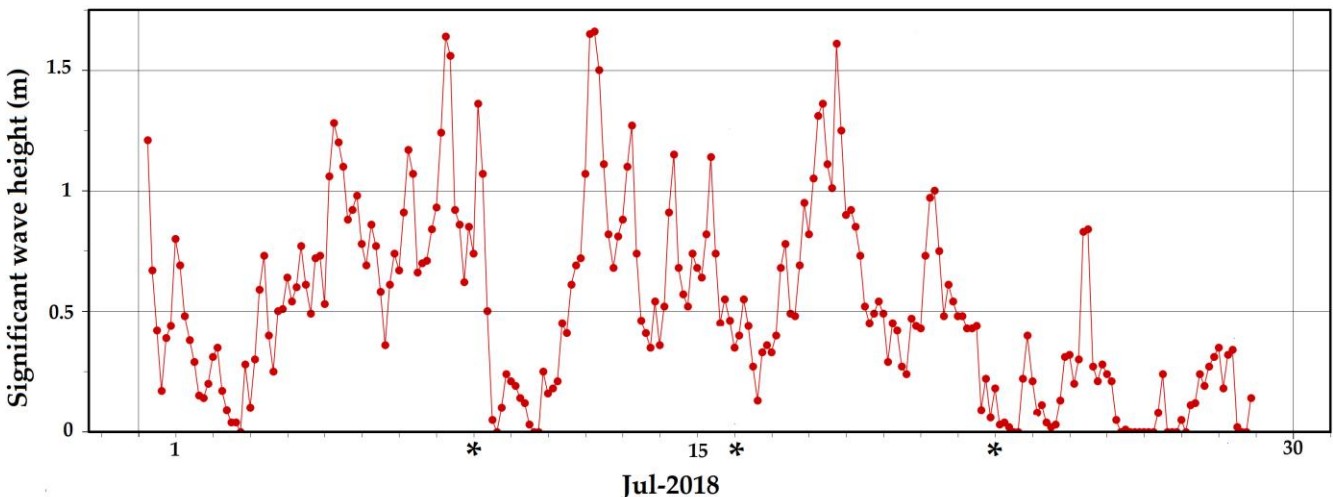

**Figure A1.** Significant wave height, which is defined as the average wave height of the highest one-third of the recorded waves, during July 2018 at investigated area. Asterisks indicate the sampling days.

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
