# Peer review of "A Preliminary Approximation to Microbiological Beach Sand Quality along the Coast of the Department of Atlántico (Caribbean Sea of Colombia): Influence of the Magdalena River"

_water, doi:10.3390/w15010048_

Round 1

Reviewer 1 Report

The question is important and there is quite a good sample size across the geographical area, but only over 2 days. Reports are clearly presented. It would be good to have a comparison or benchmark of what is considered poor bathing water quality compared to these numbers, which are quite a bit higher. 

The identification of "beach facilities" seems misleading if they are really just businesses. I see that this would attract tourists but perhaps not swimmers. It would be good to separately identify areas with facilities for beach-goers such as restrooms, parking lots, lifeguards, or changing areas if these exist anywhere on the coast. Otherwise I would suggest renaming that category "businesses on the beach." 

Is there no sewage treatment, or inadequate sewage treatment? Clarify whether septic systems or community treatment facilities exist in these locations. It would seem that prevention of contamination would be an effective approach to improve this polluted situation. 

Author Response

Dear Reviewer

Thank you so much for your very useful and constructive observations/comments, we tried to answer to all of them. All changes are marked in red in the revised version of the manuscript.

QUESTION:

The question is important and there is quite a good sample size across the geographical area, but only over 2 days. Reports are clearly presented. It would be good to have a comparison or benchmark of what is considered poor bathing water quality compared to these numbers, which are quite a bit higher. 

ANSWER:

Thank you for your observation, we added this:

Despite the above, there is presently no worldwide regulation that determines the permitted content for these microorganisms, as in the water column [23,24]. Different authors [19,21,22,25] state that studies are needed to characterize the impact that sediment-bound bacteria have on water quality and the health of bathers in order to standardize sampling and quantification methods and results such as those shown in this study seek to generate this type of discussion both in the scientific community and among environmental authorities of different countries.

QUESTION:

The identification of "beach facilities" seems misleading if they are really just businesses. I see that this would attract tourists but perhaps not swimmers. It would be good to separately identify areas with facilities for beach-goers such as restrooms, parking lots, lifeguards, or changing areas if these exist anywhere on the coast. Otherwise I would suggest renaming that category "businesses on the beach." 

ANSWER:

Thank you for your observation. Yes indeed you are right; there are not showers, toilets, etc. so we changed “facilities” for tourist activities/infrastructures that refer to restaurants, kiosks, etc.

QUESTION:

Is there no sewage treatment, or inadequate sewage treatment? Clarify whether septic systems or community treatment facilities exist in these locations. It would seem that prevention of contamination would be an effective approach to improve this polluted situation. 

ANSWER:

Thank you, we clarified this point and added few lines in the text.

"e.g. in Puerto Colombia, the largest municipality in the sampled area, only 55% of its wastewater is treated”.

Reviewer 2 Report

The manuscript deals with the analysis of sediment samples aimed to assess the role of river flow and beach uses on sediment quality. The study is carried out for the "Department of Atlántico" of Caribbean Sea of Colombia):

In general, I found the manuscript interesting, even if the results are preliminary (as clearly stated through the manuscript). 

I have some minor suggestions that the authors can accept to improve the final version of the manuscript.

#1 - Line 60: "Seawage treatment plants" are identified as a source of pollution. In my opinion, only if a poor functioning of the treatment plants applies they can be defined a source of pollution. This aspect should be clarified in the manuscript.

#2 - Line 148: A significant wave height equal to 2.0 m is declared. It is not clear which statistical measure of the wave climate the authors refer to (i.e. the mean annual value, the maximum value, etc...). This aspect should be clarified.

#3 - Line 165: Litter is commented on along with organic matter on the beaches. In my opinion, as recognized by the scientific community, organic matter (i.e. leafs, trees, trunks, etc...) can have a major role in enhancing the biodiversity of the beach system. On the other hand, plastic litter can worsen the environmental value of the beach system. I suggest clarifying this aspect in the manuscript.

#4 - Section 4: The characterization of climate characterization during the period of sampling is discussed. I suggest adding a plot showing the significant wave height time series (prior to the sampling) with the dates of sampling indicated. The same could apply also to river discharge (i.e. a water discharge time series with the dates of sampling indicated).

#5 - I suggest adding a map showing the location of sampling points in the domain of interest. 

#6 - The quality of the figures should be improved. Most of them are too small to be clear enough. 

Author Response

Dear Reviewer

Thank you so much for your very useful and constructive observations/comments, we tried to answer to all of them. All new text in the revised version of the manuscript is in red.  

QUESTION:

The manuscript deals with the analysis of sediment samples aimed to assess the role of river flow and beach uses on sediment quality. The study is carried out for the "Department of Atlántico" of Caribbean Sea of Colombia):

In general, I found the manuscript interesting, even if the results are preliminary (as clearly stated through the manuscript). 

I have some minor suggestions that the authors can accept to improve the final version of the manuscript.

#1 - Line 60: "Seawage treatment plants" are identified as a source of pollution. In my opinion, only if a poor functioning of the treatment plants applies they can be defined a source of pollution. This aspect should be clarified in the manuscript.

ANSWER:

Thank you, we clarified this point and added a few lines in the text.

QUESTION:

#2 - Line 148: A significant wave height equal to 2.0 m is declared. It is not clear which statistical measure of the wave climate the authors refer to (i.e. the mean annual value, the maximum value, etc...). This aspect should be clarified.

ANSWER:

Thank you, we added this in the text: “Significant wave height is defined as the average wave height of the highest one-third of the recorded waves.”

QUESTION:

#3 - Line 165: Litter is commented on along with organic matter on the beaches. In my opinion, as recognized by the scientific community, organic matter (i.e. leafs, trees, trunks, etc...) can have a major role in enhancing the biodiversity of the beach system. On the other hand, plastic litter can worsen the environmental value of the beach system. I suggest clarifying this aspect in the manuscript.

ANSWER:

Thank you very much for your observation, we clarified this aspect.

The abundant quantity of organic matter (i.e. leafs, branches, trees, trunks, coconuts, etc.) have a major role in enhancing the biodiversity of the beach system and, along with stranded algae, constitute nutrients for foredune pioneer plants and bushes [15]. Beach litter and, especially, plastic materials, constitute contaminant agents because of their composition and adsorbed pollutants and have a negative impact on coastal scenery [4,8,46,47]. Both organic matter and litter deposited by the Magdalena River could act as reservoirs of FIB.

QUESTION:

#4 - Section 4: The characterization of climate characterization during the period of sampling is discussed. I suggest adding a plot showing the significant wave height time series (prior to the sampling) with the dates of sampling indicated. The same could apply also to river discharge (i.e. a water discharge time series with the dates of sampling indicated).

ANSWER:

Unfortunately we have no data about river discharge but we presented in Annex 1 the required figure about daily wave climate and the date of surveys.

QUESTION:

#5 - I suggest adding a map showing the location of sampling points in the domain of interest. 

ANSWER:

Thank you. To show sampling location points we added an example of beach profile in Figure 1.

QUESTION:

#6 - The quality of the figures should be improved. Most of them are too small to be clear enough. 

ANSWER:

Thank you so much, we enlarged the size of figures as well as the size of the text within them.